# Enhanced Apiaceous Potyvirus Phylogeny, Novel Viruses, and New Country and Host Records from Sequencing *Apiaceae* Samples

**DOI:** 10.3390/plants11151951

**Published:** 2022-07-27

**Authors:** Adrian Fox, Adrian J. Gibbs, Aimee R. Fowkes, Hollie Pufal, Sam McGreig, Roger A. C. Jones, Neil Boonham, Ian P. Adams

**Affiliations:** 1Fera Science Ltd., Sand Hutton, York YO41 1LZ, UK; aimee.fowkes@fera.co.uk (A.R.F.); sam.mcgreig@fera.co.uk (S.M.); ian.adams@fera.co.uk (I.P.A.); 2Emeritus Faculty, Australian National University, Canberra, ACT 2601, Australia; adrian_j_gibbs@hotmail.com; 3School of Natural and Environmental Sciences, Newcastle University, Agriculture Building, King’s Road, Newcastle upon Tyne NE1 7RU, UK; h.l.pufal2@newcastle.ac.uk (H.P.); neil.boonham@newcastle.ac.uk (N.B.); 4UWA Institute of Agriculture, University of Western Australia, Crawley, WA 6009, Australia; roger.jones@uwa.edu.au

**Keywords:** Apiaceae, virome, *Potyvirus*, phylogeny, *Torradovirus*, *Ophiovirus*, *Umbravirus*, *Benyviridae*

## Abstract

The family Apiaceae comprises approximately 3700 species of herbaceous plants, including important crops, aromatic herbs and field weeds. Here we report a study of 10 preserved historical or recent virus samples of apiaceous plants collected in the United Kingdom (UK) import interceptions from the Mediterranean region (Egypt, Israel and Cyprus) or during surveys of Australian apiaceous crops. Seven complete new genomic sequences and one partial sequence, of the apiaceous potyviruses apium virus Y (ApVY), carrot thin leaf virus (CaTLV), carrot virus Y (CarVY) and celery mosaic virus (CeMV) were obtained. When these 7 and 16 earlier complete non-recombinant apiaceous potyvirus sequences were subjected to phylogenetic analyses, they split into 2 separate lineages: 1 containing ApVY, CeMV, CarVY and panax virus Y and the other CaTLV, ashitabi mosaic virus and konjac virus Y. Preliminary dating analysis suggested the CarVY population first diverged from CeMV and ApVY in the 17th century and CeMV from ApVY in the 18th century. They also showed the “time to most recent common ancestor” of the sampled populations to be more recent: 1997 CE, 1983 CE and 1958 CE for CarVY, CeMV and ApVY, respectively. In addition, we found a new family record for beet western yellows virus in coriander from Cyprus; a new country record for carrot torradovirus-1 and a tentative novel member of genus *Ophiovirus* as a co-infection in a carrot sample from Australia; and a novel member of the genus *Umbravirus* recovered from a sample of herb parsley from Israel.

## 1. Introduction

The family *Apiaceae* (synonym *Umbelliferae*) currently comprises 434 genera containing approximately 3700 species of herbaceous plants, characterised by hollow stems, taproots and flat topped flower heads with pedicels that radiate from a central point, called compound umbels [1] but which, in some Australian species (e.g., *Actinotus* sp.), are so contracted as to appear like single flower heads. The family contains many cultivated crop species, including carrot (*Daucus carota*), parsnip (*Pastinaca sativa*), celery (*Apium graveolens*) and arracacha (*Arracacia xanthorrhiza*); aromatic herbs such as coriander (*Coriandrum sativum*), parsley (*Petroselinum crispum*) and dill (*Anethum graveolens*); and noxious weeds such as giant hogweed (*Heracleum mantegazzianum*) and poison hemlock (*Conium maculatum*). Studies of the viruses infecting these species have only occurred sporadically, and this research has largely focused on crop species and their uncultivated relatives, which may play a role in the epidemiology of their viruses.

As the carrot is the most important crop species in the *Apiaceae*, the greatest focus has been on its viruses, with more than 30 described [2]. However, many of these viruses also infect other members of this family. Following early surveys investigating the incidence of viruses in carrot crops, research mainly focused on the twin threats of carrot necrotic dieback virus (CNDBV, Genus *Sequivirus*), formerly *Anthriscus* strain of parsnip yellow fleck virus (PYFV, Genus *Sequivirus*) [3] and the carrot motley dwarf complex (CMD) [4]. In Australasia, Europe, North America and Japan, the CMD complex comprises carrot red leaf virus (CtRLV; Genus *Polerovirus*), carrot mottle virus (CMoV; Genus *Umbravirus*) and the carrot red leaf virus-associated RNA (CtRLVaRNA) [5]. In Australia and New Zealand the *Umbravirus* carrot mottle mimic virus (CMMoV) was recognised as an alternative CMD component [6,7]. Because it occurs commonly and renders harvested carrots unmarketable, Australian research has mainly focussed on the epidemiology and the control of carrot virus Y (CarVY; *Potyvirus*) [8,9,10,11].

Within species of the *Apiaceae* other than carrot, CarVY and several potyviruses have been recorded either in single or mixed infections, and many of these have global distributions. In Australia, in addition to CarVY, apium virus Y (ApVY) and celery mosaic virus (CeMV), which occur in a range of apiaceous crops and alternative host species, have also been studied [8,10,12,13]. ApVY was reported from poison hemlock in New Zealand and from celery in association with CeMV [9]. It also occurs in the USA in a range of hosts [14] and in Europe in parsley in co-infections with carrot thin leaf virus (CaTLV) [15]. CeMV causes major losses sometimes necessitating celery free periods in Europe [16], Latin America [17], Australia [13] and especially the USA [18]. In the USA, CaTLV also infects carrot, coriander and wild apiaceous hosts [19,20]. In Europe, it also occurs in parsnip and hedge parsley (*Torilis arvensis*) [15,21,22], but it has not been reported from Australasia. Reports from the countries of the Middle East and North African are scarce. However, CarVY and the recently described gazar virus Y (GazVY) occur in Egypt, infecting carrot [23,24]. Other potyviruses, have also been found, such as angelica virus Y (AngVY) from *Angelica* spp in the NW USA [25], Japanese hornwort mosaic virus from *Cryptotaenia japonica* in Japan and *Angelica sinensis* from China [26] and poison hemlock virus Y (PHVY) from poison hemlock in Iran [27].

Reports of viruses isolated from other non-cultivated or minor apiaceous crop hosts have revealed potential cross over to other hosts. Carrot yellow leaf virus (CYLV, Genus *Closterovirus*) has been reported in hogweed (*H. sphondylium)* [28], along with several others infecting poison hemlock, including CeMV, ApVY, CaTLV and alfalfa mosaic virus (AlMV, Genus *Alfamovirus*) [9,18,29,30]. More recently, a study from the United Kingdom (UK) used high-throughput sequencing (HTS) to find the causal agent of carrot root necrosis. It revealed four previously unknown viruses in carrots, including a closterovirus (carrot closterovirus-1; CtCV-1), a torradovirus (carrot torradovirus-1; CaTV-1) and two members of a novel genus *Chordovirus* (carrot chordo viruses 1 and 2, CtChV-1 and CtChV-2). This study also associated the root necrosis symptom with CYLV [31]. Subsequently, CaTV-1 has been reported as infecting carrot in several countries including France, Germany, Greece and Japan, and also from other apiaceous hosts including celery, hedge parsley and Ashitaba (*Angelica keiskei;* a hogweed-like Japanese herb) [32,33,34,35,36]. Other studies using HTS for aetiological investigations have been difficult to interpret because they have revealed mixtures of plant viral pathogens. For example, the report of a nanovirus (parsley severe stunt associated virus) in parsley from Germany [37] was complicated by the presence of multiple alpha satellites that make it difficult to make any aetiological conclusions. Similarly, initial investigations into cilantro yellow blotch disease in the USA suggested that AlMV, beet pseudoyellows virus (BPYV; Genus *Crinivirus*) and lettuce chlorosis virus (LeCV; Genus *Crinivirus*) were involved in single or mixed infections, but they were inconclusive [38].

HTS has increasingly been used for studies of preserved historical virus samples. Sequencing plant virus isolates from historical virus isolate collections enables resolution of virus taxonomy and naming issues, and it aids dating studies. Extensive sequencing of old virus specimens avoids allocation of incorrect names to newly found viruses in metagenomic studies. With well-studied viruses, evolutionary and regional population change investigations require comparisons between old and recent isolate sequences. Historical virus collections also constitute a valuable resource enabling biosecurity investigations of virus introductions and pathways of entry and baseline surveillance. In addition, sequencing of viruses in archaeological and herbarium specimens of infected plants can also contribute to knowledge of virus evolution [39,40]. 

The aim of this study was to obtain additional information about the viruses of the *Apiaceae* by sequencing both preserved historical and recent virus specimens from different parts of the world. A special focus of this research is on potyvirus diversity.

## 2. Results

The metagenomes recovered by HTS from samples of apiaceous plants are listed in Table 1. All the recovered sequences were homologous to the genomes of viruses in the genus *Potyvirus* except those present in coriander sample 21506527 from Cyprus and Australian samples WA-1 and VIC-1 from celery and carrot, respectively. The additional metagenomes revealed in samples 21604167, WA-1 and VIC-1 had not been detected previously because the diagnostic tests used had been too specific to detect non-target viruses. All sequence data arising from the samples reported below are available in the NCBI short read archive BioProject PRJNA773642.

### 2.1. Potyvirus Detection and Diversity

First the alignments of the seven new genomic sequences, and 17 from the Genbank database related to them, were checked by RDP4 analysis [41]. Only one sequence (OK181777), that of a German CaTLV isolate, was found to be a recombinant by seven methods. These found probabilities of 10^−5^–10^−80^ of the observed anomalies being random; the sequences closest to the recombinant were CaTLV sequences, MH170889 (nts 1–6199) and LT615233 (nts 6200–10,092). This sequence was removed from further analysis.

ML analysis of the ORFs from the 7 new sequences together with those of the 16 closely related potyviruses showed that they fell into two separate lineages (Figure 1a). Most (15) were members of the previously recognised celery mosaic virus lineage [42], which also includes CarVY, ApMV and panax virus Y (PanVY). The Genbank database contains 34 additional CPs sequences of viruses in this lineage and an ML phylogeny of these together with the 23 CP sequences from the COs confirmed the groupings (Figure 1b) and added CPs of two other distinct viruses: GazVY [23] and AngVY (Robertson, 2007). The CaTLV isolates clustered with ashitabi mosaic virus (AshMV) and konjac virus Y (KonVY). The two distinct lineages were individually fully supported statistically (SH 1.0). However, there was no evidence that they are two branches of a single lineage of potyviruses because, when individual ORF sequences representing each of the viruses (ApVY, CeMV, PanVY and CaTLV) or the amino acid sequences they encode, were used for BLASTn or BLASTp searches of the Genbank database, several different potyviruses were found to be closest (i.e., there were no shared linking sequences); for example, a CeMV ORF found sweet potato virus G to be closest, CarVY found sweet potato virus 2, CaTLV found plum pox virus and using amino acid sequences CeMV found Polygonatum virus 1, CarVY found Polygonatum virus 5 and CaTLV found Ranunculus mild mosaic virus. Thus, there were no virus sequences consistently linking the two lineages, and no greater certainty was obtained using only the more conserved 3′-C-terminal halves of these sequences for these same searches.

The collection dates of 14 of the isolates providing the COs of the CeMV lineage are known. These were used with the ML phylogeny of the lineage, using the PanVY CO as an outlier, for dating analyses. The linear regression method TempEst [43] gave a possible estimate of the ‘time to most common ancestor’ (TMRCA) of 1817 CE. However, this estimate was not statistically significant (correlation coefficient 0.232; *p* = 0.425). By contrast, the ‘Relative Time Dated Tips (RTDT) relaxed clock algebraic method gave a sensible TMRCA of 1670 CE but with ‘two standard error’ dates of 1371 CE and 1824 CE and commensurate estimates for the principal nodes (Figure 2).

### 2.2. Beet Western Yellows Virus Detected from Coriander—A New Host Record in the Apiaceae

Sequence data from a sample (21506527) of coriander (Coriandrum sativum) imported to the UK from Cyprus was found to contain a sequence for the potyvirus carrot thin leaf virus (accession number OM419175). However, it was also found to contain a 5654 nt contig of a Polerovirus with 94.01% similarity to beet western yellows virus isolate SDJN16 from Capsicum annum in China (accession number MK307779.1).

### 2.3. Carrot Torrado Virus 1—A New Country Record for Australia

Sample CarVY isolate VIC-1 from a carrot in Victoria, Australia sampled in 2000–2002 was found to contain the complete genome sequence of Carrot torradovirus-1 (CaTV1; Genus *Torradovirus*). This is the first record of CaTV1 in Australia, and it represents the first record for this virus in the Southern Hemisphere. Phylogenetic trees of CaTV1 RNA 1 (Figure 3) and RNA 2 (Figure 4) confirm the taxonomic identification of the sequences and show their relatedness to other isolates of CaTV1 isolated from a range of different geographic and host sources. Phylogeny of the RNA1 (Figure 3) shows a potential minor phylogroup with the Australian and the UK carrot isolates clustered together and the CaTV1 from non-carrot hosts forming a separate minor phylogroup. However, the analysis of the RNA2 (Figure 4) indicates the two German isolates, from weeds and celery, form a minor phylogroup with isolates from Greece, Japan, Australia and the UK forming a second minor phylogroup.

### 2.4. A Novel Ophiovirus and an Unclassified RNA Virus from Australian Carrot

Sample CarVY isolate VIC-1 also contained RNA with a sequence similarity to the RNA1,2,3 and 4 of the ophiovirus Mirafiori big vein virus (MLBVV). The RNA 2 sequence contains an open reading frame encoding a putative 49 kDa protein with a 76% identity to the CP of MLBVV. The genome organisation shown in the RNA1,2,3 and 4 (OM419178–OM419181), the CP homology and the novel host (carrot) suggest that these sequences constitute a novel member of the genus *Ophiovirus* tentatively called carrot ophiovirus 1.

Additionally, sample CarVY isolate VIC-1 contained RNA consisting of 2 contigs with an identity (67%/72% respectively for the two contigs) over part (34%/−49% respectively) of their length to the unclassified and undescribed red clover RNA virus 1 (accession MG596242). Translation of open reading frames of both these contigs produced putative proteins with an identity to different parts of the replicase proteins of red clover RNA virus 1 (65%/46% identity, respectively, for the two contigs) but also Dactylorhiza hatagirea beny-like virus (65%/40% identity, respectively) (accession DAF42465) and Arceuthobium sichuanense virus 3 (65%/38% identity, respectively) (accession DAZ87284.1). This unclassified new RNA virus is tentatively called carrot associated RNA virus 1 and the sequences are available on GenBank for further study (see Table 1).

### 2.5. A Novel Umbravirus from Parsley Imported into the UK

Sample 21604167, a parsley sample imported into the UK from Israel in 2016, contained sequences of a suspected novel virus. A 3086 nt contig was obtained with little nt similarity to anything available on Genbank. The sequence contained three open reading frames (OM419177). BLAST analysis of the putative proteins showed a homology to ORF1, ORF2/RdRP and ORF3 of various umbraviruses, including Teosinte-associated umbra-like virus (accession OK018180), with a 31%, 45% and 39% identity, respectively. This homology along with the inferred genome organisation and the absence of any CP-like sequences 3′ of the ORF3 suggest that this may well be the incomplete genome of a novel umbravirus tentatively called parsley umbravirus 1.

## 3. Discussion

The metagenomes we have recovered by HTS from various samples of apiaceous plants add to knowledge of the ‘apiaceous virome’ in both breadth and depth. The largest lineage of these is of potyviruses related to CeMV, which has been described as a distinct clade [42]. This lineage now includes ApVY, CarVY and GazY from Egypt (named from the Arabic for carrot), and AngVY from two wild species *Angelica lucida* L. and *A. genuflexa* Nutt. Growing in British Colombia [25], as well as PanVY from *Panax notoginseng,* the herb ‘ginseng’, from China. All the hosts are species of the family *Apiaceae,* except for *P. notoginseng*, which is from the related family *Araliaceae*; together, they form the Apiales. The second largest lineage of potyvirus metagenomes we found are those of CaTLV, which forms a phylogenetic lineage with ashitabi mosaic virus (AshMV) found in ‘ashitabi’ (*Angelica keiskei*), a Japanese herb and a konjac virus Y (KonVY) from Chinese elephant yam (*Amorphophallu konjac*). The two lineages are individually fully supported statistically but searches of the Genbank database found no virus sequences that linked them. Similar criteria indicated that poison hemlock virus Y (PHVY) isolated in Iran from *Conium maculatum*, another apiaceous weed, is a distinct potyvirus related most closely to lettuce mosaic virus. Thus, apiaceous plants are currently hosts to at least three distinct lineages of potyviruses, which, although originally derived from a common ancestral potyvirus, probably infected non-apiaceous plant species in an intermediate ‘life’.

The COs and CPs phylogenies of the CeMV lineage differ in detail, and they give no clear indication of its ‘centre of emergence’ (CoE) [44]. The preliminary estimate of the TMRCA of the CeMV lineage as 1670 CE (2SE; 1371 CE to 1824 CE) indicated that it most likely diverged during the spread of modern agricultural practice, and the three main individual sub-lineages within it (CarVY, CeMV and ApVY) were probably spread in crop plant and weed seed. The CP phylogeny has the CeMV lineage containing viruses collected worldwide, but especially in Eurasia/North America, which is probably the center of divergence of all potyviruses [45], and also another lineage that includes many isolates of PanVY collected in China, as well as two of AngVY collected from weed Angelicas in British Colombia. It is worth noting that both the genera *Panax* and *Angelica* have classic Pacific disjunct distributions that are much more ancient [46], and the Aleut, Inuit and Yup’ik people of the region around the Bering Sea [47] used *Angelica lucida* as “a signature plant which was central to their sealing ceremonies, and sought after as a treat” [48,49]. These and similar interesting links will only be clarified when more carefully curated genomic sequences are reported from apiaceous plants.

Additionally, the samples tested here report a new host record for the *Polerovirus* beet western yellows (BWYV) in coriander (*Coriandrum sativum*) and a new country record for carrot torradovirus-1 (CaTV1) in Australia recovered from a 20-year-old carrot sample. This study also reports putative novel members of the genera *Umbravirus*, *Ophiovirus* and a virus from a potentially novel genus in the family *Benyviridae*.

Although the host range of BWYV is broad, the detection of this virus from coriander imported into the UK from Cyprus is the first record of it in a species from the family *Apiaceae* [2,50]. During the study of Adams et al. [31] “beet western yellows luteovirus ST9-associated RNA” was revealed during HTS from a carrot sample and this detection suggests a need for further work to understand the natural host range of BWYV and to investigate the role of other poleroviruses in encapsidating a range of associated RNAs.

CaTV1 was first reported in the UK and was the first member of the genus *Torradovirus* to be shown to be aphid transmitted, rather than whitefly transmitted [51,52]; but, as a recently described virus, relatively little is known about its biology, distribution or diversity. With the detection and the molecular characterisation of further isolates from Europe and Japan infecting a range of apiaceous hosts [32,33,34,35,53], the distribution and the diversity of this virus is gradually being revealed. The isolate “CaTV1-celery” from Germany was reported to show distinct differences in nt and amino acid sequence and a difference in the size of RNA1 and RNA2 [34] as compared to type isolate H6 [31]. Data presented for CaTV1 RNA1 suggest that there are potentially two clades of CaTV1, with one clade encompassing the carrot isolates from the UK and from Australia, and a second clade with the non-carrot isolates including a weed and celery from Germany and *A. keiskei* from Japan, although the *Anglica* isolate appears to be basal to this clade. Unfortunately, the complete RNA 1 sequence for the Japanese carrot isolate reported by Yoshida [53] and the Torillis isolate from [33] were not available for this analysis. The analysis of the available complete RNA 2 sequence again appears to cluster with the two German isolates, from celery and an unidentified weed, together. However, a *Torillis* isolate from Greece lineages with the two carrot isolates (UK and Australia), and the RNA 2 of the Angelica isolate is basal to this clade rather than the German/non-carrot clade. Given there are still relatively few isolate sequences of CaTV1 available, surveillance work should be carried out to further investigate the diversity and the distribution of this emerging species. The reports highlighting the global distribution of recently described viruses, such as CaTV1, and the close relationship between minor phylogroups of these isolates (Figure 3 and Figure 4) may also indicate that the virus has a seed borne-dissemination route, enabling global movement of the virus. However, this would require further investigation.

In addition to the unexpected findings of known viruses, a tentative novel virus from the genus *Ophiovirus* was revealed in the same 20-year-old Australian carrot sample as the sequence of carrot torradovirus-1. Although, ophioviruses infect a broad range of plant taxa, this appears to be a first record of a virus from this genus infecting a member of the *Apiaceae*. Relatively little work has been conducted on transmission of the viruses of this genus, but they can be mechanically and vegetatively transmitted to a limited range of test plants. Of the seven ICTV ratified species in the genus, four are reported to be transmitted via soil-borne fungus of the genus *Olpidium*. Further work is therefore needed to biologically and molecularly characterise this virus and to ascertain its distribution, impact and epidemiology in Australia and elsewhere. Due to a lack of knowledge of symptomology in this specific sample with co-infecting viruses, no inferences can be drawn on any disease associated with it [54], as a result we propose the virus name carrot ophiovirus-1 (CtOV1) with a proposed species name of *Ophiovirus daucus*.

The 20-year-old carrot sample from Australia found to contain CaTV-1 and CtOV1 also produced sequenced data consisting of two contigs with homology to a range of viruses. Detailed analysis of two of these, Dactylorhiza hatagirea beny-like virus and Arceuthobium sichuanense virus 3, suggested they form part of a “sister clade” of plant infecting viruses along with Red Clover virus 1 also related to benyviruses and in the family *Benyviridae* [55]. Based on its sequence identity, the virus in the Australia carrot may form a fourth member of this lineage, which may form a novel genus related to the genus *Benyvirus* in the family *Benyviridae*. As above, due to co-infecting viruses and the lack of any symptom record, no conclusions can be drawn about the symptoms of this virus, and thus we suggest the tentative name of carrot associated RNA virus 1. Further work would be required to confirm the status of this virus as a member of a novel genus within the family *Benyviridae*.

The detection of a tentative novel member of the genus *Umbravirus* was inferred from a sample of parsley plants imported to the UK from Israel in 2016. The sample was intercepted by import inspectors due to the plant exhibiting chlorosis, consistent with possible viral infection (Figure 5b). However, the sample had tested negative using a range of PCR primers to apiaceous viruses and phytoplasmas [32,52,56] and so it was subsequently selected for HTS to rationalise the causal agent of the observed symptoms. It is notable that the virus was detected as a single infection. Under natural conditions umbraviruses are traditionally thought to require the presence of a specific helper virus from the family *Solemoviridae* (formerly Luteoviridae) [56]. There are increasing reports of umbraviruses or umbra-like associated RNAs [57,58,59] being found either alone or without a luteovirus helper virus. As umbraviruses do not produce a CP and are reliant on their helper virus for aphid transmission, there are yet unresolved questions on how they spread in the absence of a helper [60]. This could be because the host was resistant to the helper virus therefore leaving the umbravirus as a lone infection [61,62], the umbravirus was being mechanically transmitted [60], or possibly another mechanism yet to be elucidated. Possibly, it had entered the plant via mechanical inoculation, e.g., during plant propagation. Although no other virus was detected in the sample, it cannot be assumed that the observed symptom is a result of viral infection without further work being conducted to investigate this relationship [54], as a result we propose the virus name parsley umbravirus 1 (PaUV-1), with a proposed species name of *Umbravirus petroselinum*.

Whilst reporting novel sequences to further elucidate the diversity of potyviruses infecting apiaceous species, this study also highlights the remaining knowledge gaps in the evolutionary history of this key group of viruses infecting such a globally important plant family, which includes crops and ecologically important uncultivated species, as mentioned in the introduction. Furthermore, the additional new host record (BWYV in Coriander), new country record (CaTV1 in carrot from Australia) as well as three putative novel viruses further emphasises the need for a greater focus on the *Apiaceae* virome to understand the pathological and the ecological relationships between its viruses and their respective hosts.

## 4. Materials and Methods

### 4.1. Sample Source

Preserved samples were from both historical and recent sample collections that lacked sequence data. These represented a range of apiaceous crop and herb species drawn from two sources, as detailed in Table 2. Four samples were taken from UK plant health import interceptions between 2015–2018, consisting of either coriander or parsley originating from the Mediterranean region (Cyprus, Israel and Egypt). These samples had been tested by qPCR for the presence of phytoplasma [63] and qRT-PCR for a range of carrot infecting viruses including CtRLV, CMoV, CaTV-1, CtCV-1 and CYLV [31,52], but the results were negative in both studies. Consequently, the samples were retested using HTS to determine the identity of any other viruses that might be present. A further six samples were collected in Australia during surveys for viruses of apiaceous crops in 1998-2002. These had previously been found to contain the potyviruses ApVY, CeMV or CarVY identified through specific serological testing by ELISA, sometimes accompanied by inoculation to test plants but not by qPCR or sequencing [8,13].

### 4.2. High Throughput Sequencing (HTS) and Bioinformatics

Samples of fresh and previously freeze-dried infected leaf material (Table 2) were subjected to HTS. Briefly, total RNA was extracted from each sample using a Total RNA kit (Qiagen, Manchester, UK), including DNAase treatment. Indexed sequencing libraries from the four UK import interception samples were produced using the Scriptseq complete plant leaf kit (Illumina, San Diego, CA, USA), following manufacturer’s instructions and as described in [31]. Indexed sequencing from the six Australian survey samples were produced using the TruSeq complete plant leaf kit (Illumina, San Diego, CA, USA), following the manufacturer’s instructions and as described in [64].

The indexed libraries were sequenced along with others on a MiSeq (Illumina), using a 600 cycle V3 kit. The resulting paired end reads were 3′ trimmed to a quality score of 20 using Sickle in paired-end mode [65]. Contigs were assembled using Trinity v2 [66]. BLAST+ was employed to compare the contigs obtained to the Genbank non-redundant (nr) and nucleotide (nt) databases [67]. MEGAN Community Edition [68] was used to extract reads of viral origin.

To calculate the read number and the coverage, raw reads were mapped back to the viral genomes identified using BWA-MEM2 [69]. Subsequent files were processed with SAMtools [70], and coverage was generated using the BamToCov software [71].

### 4.3. Phylogenetic Analyses

The Genbank nt database was searched, using its BLAST facility, for all sequences similar to those of the 7 new potyvirus genomic sequences reported in this paper (Table 1) and 17 additional complete genomic sequences were found on Genbank. All 24 sequences were edited using BioEdit [72] to extract the complete genomic open reading frame sequences (COs) or their coat protein (CP) genes. These were aligned, using the encoded amino acids as a guide, by the TranslatorX online server (http://translatorx.co.uk (accessed on 3 March 2022)) [73], with its MAFFT option [74].

The appropriate substitution model for evolutionary analysis of nt sequences was determined using MEGA-11 [75], and, as a result, the phylogenies of nt alignments were inferred using the GTR + γ + I substitution model. Sequences, both COs (23) and CPs (57, including 34 more from Genbank), were tested for the presence of phylogenetic anomalies using the full suite of options in RDP4 [41], with default parameters [41,76,77,78,79,80,81,82,83,84,85]. The most appropriate substitution model for phylogenetic trees was calculated using the maximum likelihood algorithm in Mega 7 [86] or in PhyML 3.0 [87], and node support was assessed using 500 bootstraps in Mega-7 or the Shimodaira–Hasagawa (SH) statistics in PhyML. Trees were drawn using not.3 (http://tree.bio.ed.ac.uk/software/figtree/, accessed on 12 May 2018) and a commercial graphics package. The phylogenetic position of the partial GazVY CP gene sequence was determined by aligning it with the other CP gene sequences and using only the region found in all sequences for the ML analysis.

The 15 COs for which collection dates were known were analysed for their ‘temporal signal’ by the linear regression method TempEst [43] and the relaxed clock algebraic method RTDT (Relative Time Dated Tips; [88]), which is an option in MEGA-11.

## Figures and Tables

**Figure 1 plants-11-01951-f001:**
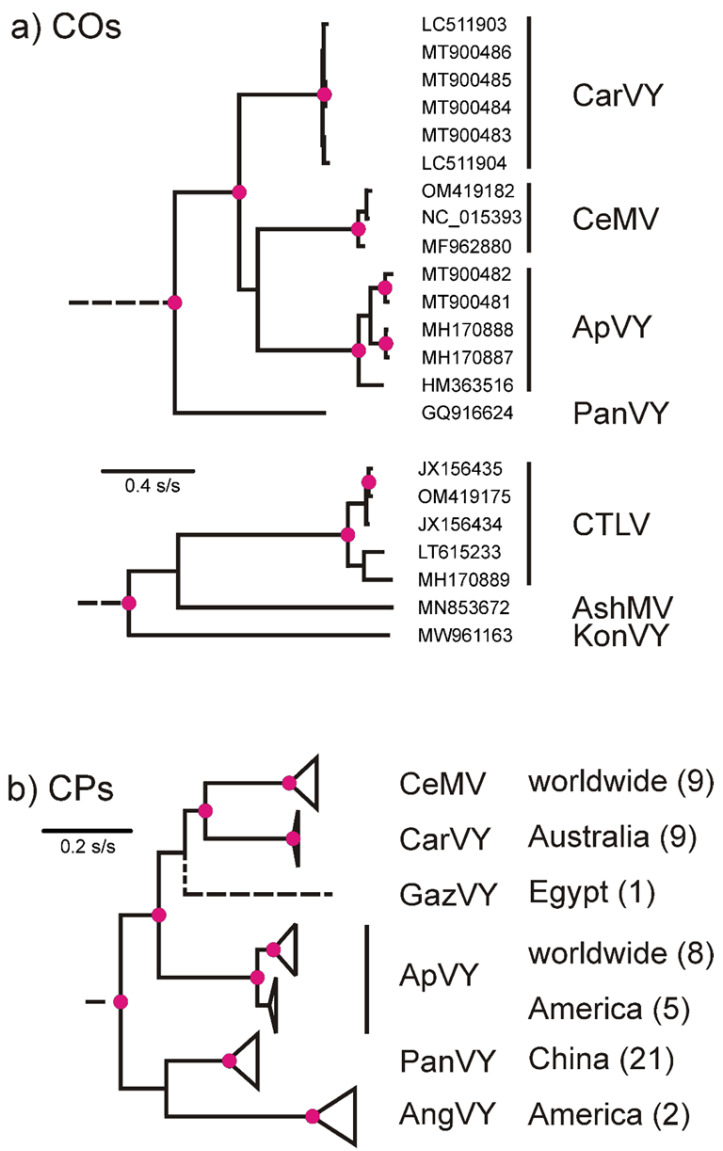
ML phylogenies of: (**a**) the non-recombinant complete ORFS (COs) of the two lineages of potyviruses that were analysed; (**b**) the coat protein (CP) genes from the COs together with those downloaded from Genbank; the total number of sequences contributing to each ‘collapsed’ cluster in parentheses. Abbreviations: AngVY, angelica virus Y; ApVY, apium virus Y; AshMV, ashitabi mosaic virus; CarVY, carrot virus Y; CeMV, celery mosaic virus; CaTLV, carrot thin leaf virus; GazVY, gazar virus Y; KonVY, konjac virus Y; PanVY, panax virus Y. The partial ORF from sample 21625530, was of nts 928–1249 and 5860–6309 of the other sequences, and when these regions were compared, they showed the isolate to be closest to MT900482. The Gazar virus Y partial CP gene (GQ148776) was of 320 nts from the centre of the CP gene, and it was placed in the phylogeny using a phylogeny calculated using only the homologous region of all 80 other CP genes. The other CP genes are: AngVY, EF488740 and NC_043138; ApVY-Am, EU255632, EU515125, EU515126, FJ010828 and HM363516; ApVY-W, AF203529, AF207594, AY049716, FJ010827, MH170887, MH170888, WAC10577, and 21806162; CarVY, AF203538, AF203539, LC511903, LC511904, NC_043142, WAC10143, WAC10144, WAC10145 and WAC12419; CeMV, AF203531, AF203532, AF203533, AF203534, AF203535, AJ271087, HQ676607, MF962880, MK570304 and NC_015393; and for PanVY, DQ452932, FJ816101, GQ916624, JX106467, JX106468, JX106469, JX106470, JX106471, JX106472, JX106473, KF150748, KF150749, KF150750, KF150751, KF150752, KF150753, KF150754, KF150755, KF150756, KF150757 and KF150758. Red disks are on nodes with >0.99 SH support.

**Figure 2 plants-11-01951-f002:**
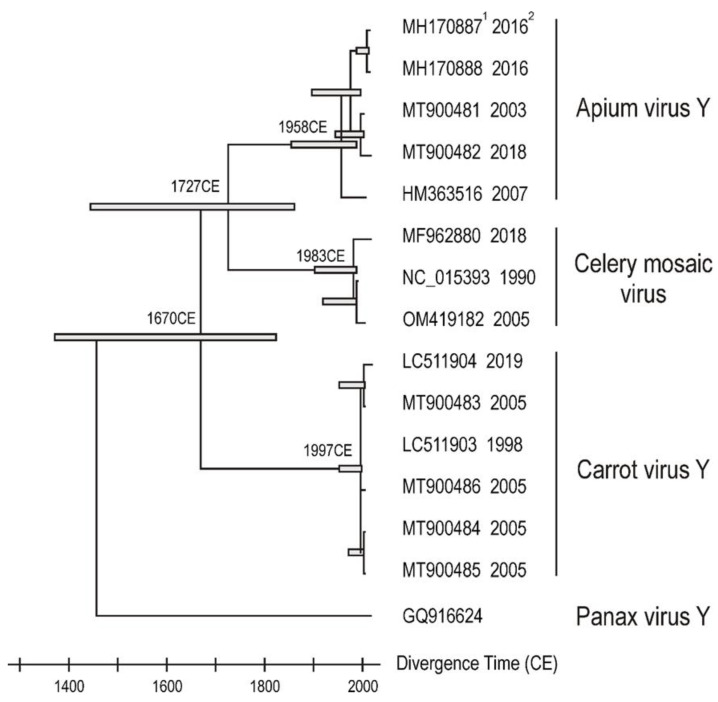
A dated phylogeny of the 14 CeMV lineage sequences for which there are known ‘collection dates and complete genome sequences. The accession codes of the sequences (1), and the collection dates of the isolates (2) are shown. The analysis was done using the RTDT method in Mega 11. The error bars for the major nodes are two standard errors.

**Figure 3 plants-11-01951-f003:**
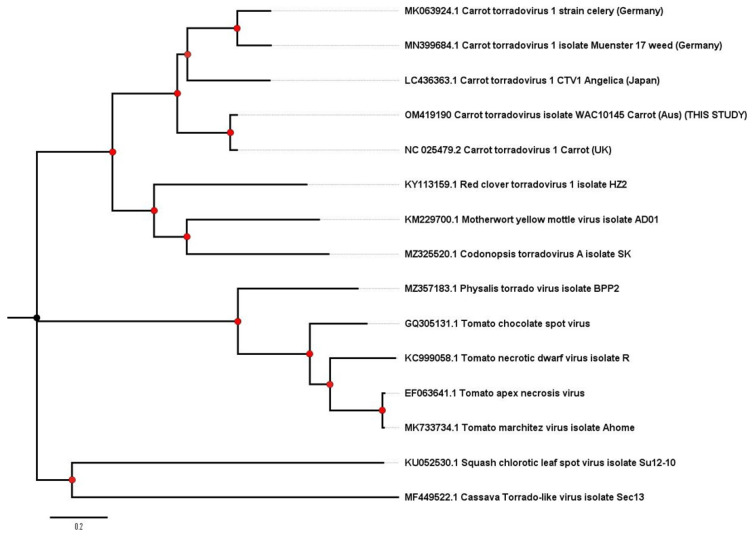
Phylogenetic tree of complete RNA1 nucleotide sequences of CaTV1 and a range of related torradoviruses. For the CaTV1 isolates, host and country of isolation are included.

**Figure 4 plants-11-01951-f004:**
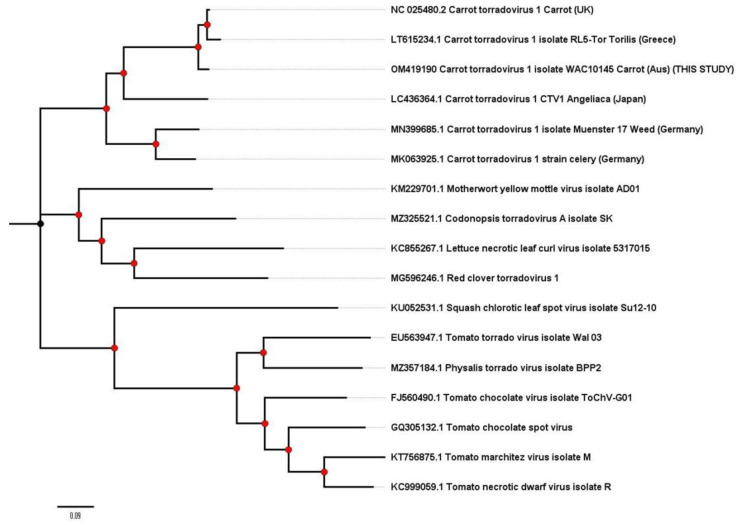
Phylogenetic tree of complete RNA2 nucleotide sequences of CaTV1 and a range of related torradoviruses. For the CaTV1 isolates, host and country of isolation are included.

**Figure 5 plants-11-01951-f005:**
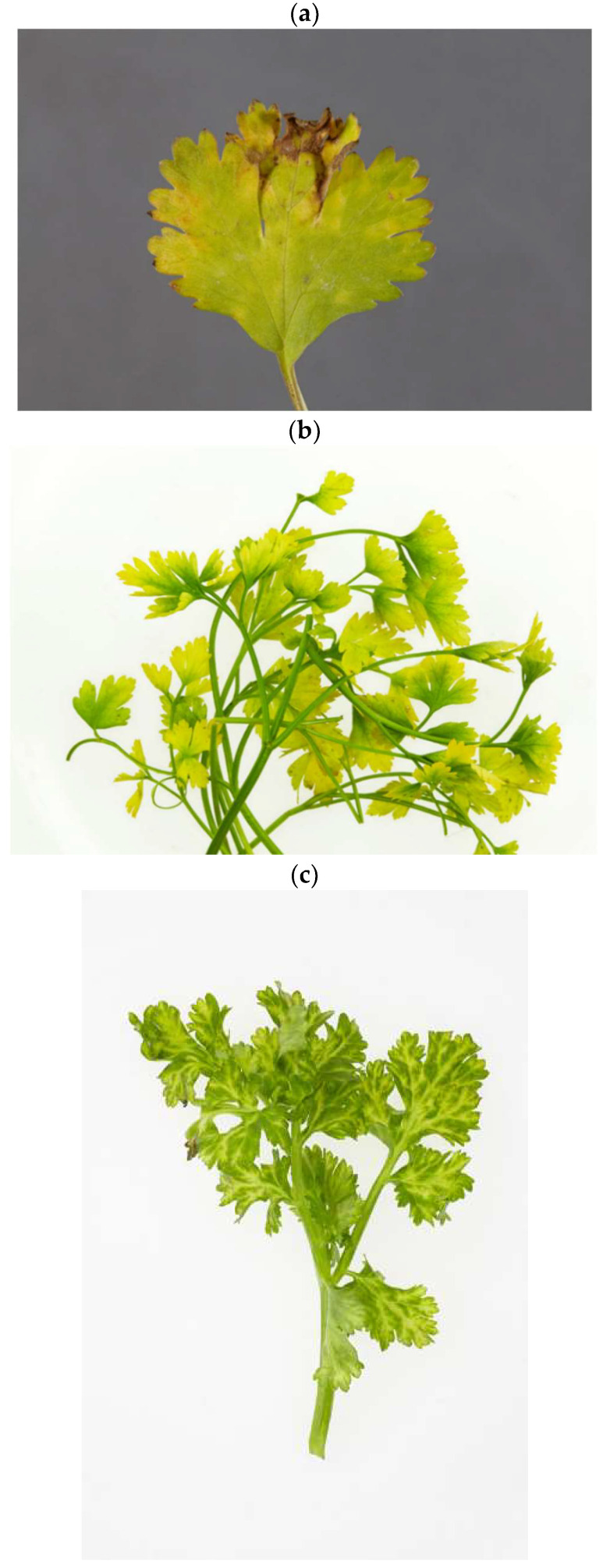
Photographs of symptoms in leaves from apiaceous samples intercepted by UK plant health and seeds inspectors; (**a**) Sample 21506527, coriander showing chlorotic spotting; (**b**) Sample 21604167, parsley showing general chlorosis and chlorotic mottle; (**c**) Sample 21625530, coriander showing vein yellowing and chlorotic mottle; (**d**) Sample 21806162, coriander showing vein yellowing and chlorotic spotting.

**Table 1 plants-11-01951-t001:** Metagenomes of previously unsequenced viruses obtained from preserved historical samples from apiaceous plants. * indicates tentative virus name for a first record, ** indicates a new host record, and *** indicates a new country record.

Sample ID	Host (Latin/Common Name)	Paired End Reads	Virus Present	Coverage	Accession Code
21506527	*Coriandrum sativum* Coriander	745985	carrot thin leaf virus beet western yellows **	11686.84 231.82	OM419175 OM419176
21604167	*Petroselinum crispum* Parsley	1092446	parsley umbravirus 1 *	428.83	OM419177
21625530	*Coriandrum sativum* Coriander	5823	apium virus Y	0.52	OM419183-OM419187 (partial sequence)
21806162	*Coriandrum sativum* Coriander	147561	apium virus Y	1532.98	MT900482
ApVY isolate WA-1	*Petroselinum crispum* Parsley	1083897	apium virus Y	649.53	MT900481
CeMV isolate WA-1	*Apium graveolens* Celery	1629326	celery mosaic virus	628.97	OM419182
CarVY isolate SA-2	*Daucus carota* Carrot	1172179	carrot virus Y	7198.74	MT900484
CarVY isolate SA-3	*Daucus carota* Carrot	918475	carrot virus Y	6467.35	MT900486
CarVY isolate SA-4	*Daucus carota* Carrot	969303	carrot virus Y	2725.14	MT900485
CarVY isolate VIC-1	*Daucus carota* Carrot	1105338	carrot virus Y carrot torrado virus-1 *** carrot ophiovirus-1 * Carrot associated RNA virus 1	2883.78 1080.96, 1534.21 13.29, 87.48, 129.41, 45.37 3.71, 4.67	MT900483 OM419190-OM419191 OM419178-OM419181 OM419188-OM419188

**Table 2 plants-11-01951-t002:** Origin of samples and sample descriptions.

Sample ID	Host (Latin/Common Name)	Sample Description	Origin	Symptoms
21506527	*Coriandrum sativum* Coriander	Frozen RNA extract from fresh leaf	UK import interception Ex. Cyprus, 2015	Chlorotic spotting See Figure 5a.
21604167	*Petroselinum crispum* Parsley	Frozen RNA extract from fresh leaf	UK Import interception Ex. Israel 2016	General chlorosis and chlorotic mottle See Figure 5b.
21625530	*Coriandrum sativum* Coriander	Frozen RNA extract from fresh leaf	UK Import interception Ex. Egypt 2016	Vein yellowing and chlorotic mottle See Figure 5c.
21806162	*Coriandrum sativum* Coriander	Frozen RNA extract from fresh leaf	UK Import interception Ex. Egypt 2018	Vein yellowing and chlorotic spotting See Figure 5d.
ApVY isolate WA-1	*Petroselinum crispum* Parsley	Freeze-dried leaf	Survey sample Ex. Perth metropolitan area, south-west Australia, 2002 [10]	Mosaic
CeMV isolate WA-1	*Apium gravelolens* Celery	Freeze dried leaf	Survey sample Ex. Perth metropolitan area, south-west Australia, 1998 [13]	Mosaic
CarVY isolate SA-2	*Daucus carota* Carrot	Freeze dried leaf	Survey sample Ex. South Australia, 2000–2002, [9]	Unknown, from random sample
CarVY isolate SA-3	*Daucus carota* Carrot	Freeze dried leaf	Survey sample Ex. South Australia, 2000–2002 [9]	Unknown, from random sample
CarVY isolate SA-4	*Daucus carota* Carrot	Freeze dried leaf	Survey sample Ex. South Australia, 2000–2002 [9]	Unknown, random sample
CarVY isolate VIC-1	*Daucus carota* Carrot	Freeze dried leaf	Survey sample Ex. Victoria, Australia, 2000–2002 [9]	Unknown, from random sample

## Data Availability

Data available in a publicly accessible repository. All sequence data arising from the samples reported are available in the NCBI short read archive BioProject PRJNA773642. Sequence data from individual viruses reported here are available in NCBI GenBank under the accession codes listed in Table 1.

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
