# Peer review of "Enhanced Apiaceous Potyvirus Phylogeny, Novel Viruses, and New Country and Host Records from Sequencing Apiaceae Samples"

_plants, 2022, doi:10.3390/plants11151951_

Round 1
Reviewer 1 Report
Since numerical citation has been used throughout the manuscript there is no need to include the author names,Thus, on line 132 replace "Martin et al 2015" with a numerical citation. In line 252 delete "Wu, Liu, Liu, Li and Li" On line 302 delete"by Gaafer and Ziebelle". On line 418 delete "in Adam et al". On line 421 delete "in Fox et al"
Reviewer 2 Report
The paper "Enhanced apiaceous potyvirus phylogeny, novel viruses, and new country and host records from sequencing Apiaceae samples" reported the high throughtput sequencing of historical and recent Apiaceae samples. The research is interesting and well conducted the results clear and all the paper carfully written and revised by the authors.
In my opinion only two information lack from the article:
1. Are the sample been previuosly tested from the viruses retrieved?
2. AAs should include some information (i.e. as table in the supplementary materials) about the quality of the NGS sequence, like the coverage etc..
